# Is Promotion of Mobility in Older Patients Hospitalized for Medical Illness a Physician’s Job?—An Interview Study with Physicians in Denmark

**DOI:** 10.3390/geriatrics5040074

**Published:** 2020-10-10

**Authors:** Mette Merete Pedersen, Rasmus Brødsgaard, Per Nilsen, Jeanette Wassar Kirk

**Affiliations:** 1Department of Clinical Research, Copenhagen University Hospital Hvidovre, Kettegaard Allé 30, 2650 Hvidovre, Denmark; rasmusbroed@gmail.com (R.B.); Jeanette.Wassar.Kirk@regionh.dk (J.W.K.); 2Department of Clinical Medicine, University of Copenhagen, 2200 Copenhagen, Denmark; 3Department of Health, Medical and Caring Sciences, Linköping University, 58183 Linköping, Sweden; per.nilsen@liu.se; 4Department of Public Health, Nursing, Aarhus University, 8100 Aarhus, Denmark

**Keywords:** mobility intervention, hospitalization, older medical patients, physician perspective, barriers and facilitators

## Abstract

The aim of this study was to identify the most common barriers and facilitators physicians perceive regarding their role in the promotion of mobility in older adults hospitalized for medical illness as part of on an intervention to promote mobility. Twelve physicians at two medical departments were interviewed face-to-face using semi-structed interviews based on the Theoretical Domains Framework. The physicians’ perceived barriers to promoting mobility were: the patients being too ill, the department’s interior does not fit with mobility, a culture of bedrest, mobility not being part their job, lack of time and resources and unwillingness to accept an extra workload. The facilitators for encouraging mobility were enhanced cross-professional cooperation focusing on mobility, physician encouragement of mobility and patient independence in e.g., picking up beverages and clothes. The identified barriers and facilitators reflected both individual and social influences on physicians’ behaviors to achieve increased mobility in hospitalized older medical patients and suggest that targeting multiple levels is necessary to influence physicians’ propensity to promote mobility.

## 1. Introduction

In older adults (>65 years), low levels of mobility and excessive bed rest during hospitalization have been linked with decreased independency after discharge, institutionalization, readmission and death [1,2,3,4,5,6,7,8]. As argued in a recent editorial [9], hospital associated disabilities are still common despite three decades of study and knowledge about modifiable factors during hospitalization—e.g., limited mobility. Several studies have reported low levels of mobility in older patients hospitalized for medical illness (henceforth: older medical patients) [10,11,12,13,14]. Additionally, in a previous study, we found that older medical patients spent a median of 22 h a day lying or sitting and less than one hour a day standing and walking during hospitalization [15].

Interventions seeking to increase mobility in older medical patients may have the potential to improve function, shorten length of stay in the hospital and lower institutionalization rates [16,17,18]. However, maintaining a culture that encourages such interventions could prove both time and resource consuming and requires a well-organized, interprofessional approach [19].

To increase the likelihood of successful implementation of a mobility intervention, it can be beneficial to tailor the intervention to the context and take all relevant barriers towards change into account [20]. Furthermore, tailoring an intervention may increase the chances that the implementation of the intervention is sustained once the project has ended and the researchers have left [21]. Finally, despite evidence of methodological inconsistencies and varying effect evaluation in the published literature, adopting a user engagement strategy in the design process of interventions is becoming increasingly common [22,23,24]. User engagement is thought to make research more relevant for stakeholders and increase the likelihood of successful implementation [22,25].

The present study is part of the WALK-Copenhagen project (WALK-Cph), which was designed employing a user engagement strategy and thereby tailored to its context and stakeholders. The methods of WALK-Cph have been described in detail elsewhere [26]. Briefly, WALK-Cph is a pragmatic cross-sectoral project aiming at increasing mobility during hospitalization and following discharge in older patients admitted for acute medical illness. Using both qualitative and quantitative methods, WALK-Cph is based on a hybrid-design to simultaneously investigate the clinical effect and the fidelity and adoption of an intervention to increase mobility and its implementation [27]. The intervention and its implementation plan were developed through a series of workshops in collaboration with key stakeholders from two departments at two different hospitals (physicians, trained nurses, nursing assistants, physiotherapists, occupational therapists, patients and their relatives) [25] and thereby tailored to the local context [20]. The mobility intervention, which was developed in the workshops, consists of several components: on admission, patients are given a welcome folder with information on the importance of being mobile during hospitalization; posters on the walls encourage patients to walk and exercise; staff are to encourage patients to use a WALK-path placed in the hallway and to self-service on beverages and clothes; physicians are to prescribe WALK-plans to all patients capable of walking and encourage all patients to be physically active during hospitalization; and after discharge, municipality staff are to follow-up on WALK-plans. However, we experienced limited physician involvement in the design process since only one physician attended the workshops. This left us with inadequate knowledge about barriers and facilitators (i.e., determinants) experienced and/or perceived by physicians regarding the intervention to promote mobility as well as their own promotion of mobility in older medical patients.

In several studies, a physician’s advice has been shown to be effective in promoting exercise among older adults in primary care [28,29,30]. Additionally, hospitalized older medical patients have underlined the importance of a health care professional, and preferably a physician, telling them to be active during hospitalization [31,32]. However, a systematic literature search on barriers and facilitators concerning physicians’ promotion of mobility in their older medical patients revealed a lack of research on the subject. Only few studies have investigated physicians experiencing barriers towards mobility [31,33]. Therefore, the aim of this study was to classify the most common barriers and facilitators to physicians’ promotion of mobility in older medical patients as part of an intervention to promote mobility using the Theoretical Domains Framework.

## 2. Materials and Methods

### 2.1. Study Design

#### 2.1.1. Qualitative Approach

This was a qualitative study using face-to-face semi-structed interviews [34] to explore barriers and facilitators to physicians’ promotion of mobility in the WALK-Cph mobility intervention. The interviews were conducted following the WALK-Cph design workshops and before feasibility and fidelity testing of the intervention [26]. After the design workshops, the WALK-Cph intervention consisted of seven components as mentioned previously.

#### 2.1.2. Semi-Structured Interviews

Theoretical grounding of implementing interventions to influence behavior in health care professionals is important to assist in understanding and explaining how and why implementation succeeds (or fails) [35]. The Theoretical Domains Framework (TDF) is a widely used and validated framework that categorizes different types of influences on behavior change (i.e., domains). TDF contains 128 different psychological constructs derived from 33 behavior change theories and clustered into 14 independent domains [36,37]. It guides researchers on how to identify and categorize factors that either prevent (i.e., barriers) or aid (i.e., facilitators) a certain behavior. The TDF has demonstrated to be a useful implementation framework for health care professionals in assessing barriers and facilitators that influence behavior change [38]. Atkins et al. presented a guide to using the TDF to identify influences—e.g., barriers and facilitators, relevant to the implementation of a certain behavior [39]. Aided by this guide and a previous ethnographic field study conducted in the two medical departments [19], we developed an interview guide for the semi-structured interviews based on the TDF domains. The target behavior in this study was physicians’ promotion of mobility in hospitalized older medical patients as part of the WALK-Cph intervention and thus, the questions were focused on obtaining information about the barriers and facilitators perceived by physicians towards this behavior. Additionally, introductory information was obtained on the physicians’ promotion of mobility in hospitalized older medical patients in general to direct their focus on the subject.

Between one and five open questions were developed to address each domain of the TDF, adding to a total of 29 questions (Appendix A). All informants received the same questions. No specific questions were developed for the emotion domain. We expected that information about emotions would be implicitly covered and that specific questions would seem irrelevant in this interview regarding promotion of mobility. According to the semi-structured interview method, additional follow-up questions targeting a specific domain were allowed if elaboration of a response was needed [34]. Since the physicians had declined to attend the initial design workshops and, therefore, had not had the opportunity to contribute to the development of the intervention, additional questions were added to the interview guide in order to explore the physicians’ perceptions of the intervention’s feasibility as well as their own ideas about the contents of an intervention to enhance mobility.

Prior to data collection, the interview guide was pilot tested with a non-affiliated geriatrician. This resulted in changes in the wording of some questions to improve understandability and removal of certain questions that appeared to be redundant (please see Appendix A for the Interview Guide).

### 2.2. Setting

The study was carried out in Denmark, where the health care system is publicly funded from taxes. The Danish welfare state provides free treatment for primary medical care, hospitals and home-based care services for all citizens. As mentioned, the intervention of the WALK-Cph project took place in two medical departments (X and Y) at two hospitals (X and Y) in the Capital Region of Denmark. Most patients admitted to the two intervention departments are acutely admitted older medical patients. The two departments are similar in size and staff composition. Department X has 24 beds and 36 staff members (18 nurses, 6 certified nursing assistants and 12 physicians) with responsibilities in the department. Department Y has 25 beds and 37 staff members (18 nurses, 3 certified nursing assistants and 8 physicians) with responsibilities in the department). All physicians in the two departments are either geriatricians or medical physicians, who provide hospital-based care. At the two hospitals, the therapists are organized differently and have different links with the departments. At Hospital X, the therapists are organized into a separate department and visit Department X to attend to referred patients. At Hospital Y, three physiotherapists are a part of the Department Y staff. 

### 2.3. Sampling Strategy

We used a purposive sampling strategy with the criterion that all participants were physicians from one of the two intervention departments [40]. We aimed for maximum variation based on age, sex, years of working experience and medical specialty (geriatric or general medical) to make sure that all aspects of the phenomenon were investigated as comprehensibly as possible within the group and to interview as diverse a population as possible within the two departments’ physicians [41]. Participants were invited via e-mail. Additional e-mails were sent to those who did not reply to the initial e-mail. Additionally, two representatives from the WALK-Cph research group (J.W.K. and a chief physician) attended a morning meeting at each department to inform the physicians about the WALK-Cph project and describe the interviews and e-mail invitations. At the following morning meeting, the head physicians in both departments reminded the physicians to reply to the e-mails, either with a confirmation or rejection to participate. Sampling ended when all invited physicians had either accepted or declined the invitation.

### 2.4. Data Collection

Two researchers from the research group (J.W.K. and M.M.P.) performed 12 interviews between February and June 2018. Both had previous interviewing experience and a comprehensive understanding of the target behavior. The interviews were conducted in the participating physician’s department in an undisturbed office. Each interview lasted approximately 30 to 45 min and was conducted in Danish. The interviewers conducted the first two interviews together to provide mutual feedback and thereby ensure consistency in interviewing. The remaining 10 interviews were conducted individually. The interviews were audio-recorded and transcribed verbatim after each interview by one of two research assistants. Throughout the data collection period, transcriptions were read by the interviewers to identify inadequate domain coverage so that the interview guide could be refined if necessary. Fortunately, this was not necessary. The transcripts of the interviews were assigned with a code number to allow tracking and development of a coding approach.

### 2.5. Data Analysis

We conducted a qualitative directed content analysis with a focus on manifest content [42,43,44]. According to Joffe and Yardley, manifest content refers to what is directly observable and visible in the data [43]. Three researchers (J.W.K., M.M.P. and R.B.) carried out the analysis. One of these researchers (J.W.K.) had previous experience in using the TDF. with another researcher (P.N.) [45], who acted as a supervisor during the analysis process to ensure that the application of the TDF was satisfactory. The aim of the analysis was to identify the most common barriers and facilitators among the physicians concerning the WALK-Cph mobility intervention and their own promotion of in-hospital mobility. We defined barriers as physicians’ negative attitudes and behaviors in relation to the intervention and/or their promotion of mobility, while facilitators were defined as physicians’ positive attitudes and behaviors concerning the intervention and/or their promotion of mobility.

We analyzed the interviews using a directed content analysis as described by Hsieh and Shannon [44] and deductively analyzed the transcribed interviews using an iterative process based on the TDF [39]. Firstly, we agreed on a coding approach to ensure consistency in the coding process. This approach encompassed that coding of the transcripts should be performed by reading the transcripts and searching for specific words or statements that were considered to be related to any of the contents of the 14 TDF domains, as described by Atkins et al. [39]. Each domain of the TDF was considered a coding category [44]. Secondly, before analyzing the transcribed data, the coders examined the domains of the TDF and agreed on how to interpret and apply each domain. For example, it was agreed that the optimism domain would refer to the physicians’ expressed own optimism/pessimism and not that of other health care professionals or patients. It was also agreed that statements referring to aspects of physical time would be coded under the “Environmental Context and Resources” domain. After agreeing upon this coding approach, two of the researchers (J.W.K. and M.M.P.) independently coded the first two interview transcripts. Each coder independently read each interview and identified and highlighted meaning units (i.e., statements) that were related to the target behavior. Then, the two coders re-read the interviews and considered if any meaning units (i.e., “a constellation of words or statements that relate to the same central meaning” [46]) that were related to the target behavior were of relevance to any of the TDF domains. Meaning units of relevance to the TDF domains were then allocated to the relevant domain [39]. After independent coding of the interviews, the allocations of meaning units were discussed between all three coders, and possible differences were considered until consensus was reached, after which the coding approach was refined [39]. This process (i.e., coding, consensus, refinement) was used for all interviews. The subsequent three interviews were coded independently by all three coders, whereas the remaining seven interviews were coded by two coders (M.M.P. and R.B.). After the initial coding of all interviews, differences in meaning unit allocations were discussed by all three researchers until negotiated consensus was reached on distribution of all meaning units. When a meaning unit was relevant to more than one domain of the TDF, it was coded into several domains. In some cases, all assigned domains for one meaning unit were agreed upon, and in other cases, negotiated consensus was reached on de-assigning a domain to a given meaning unit. The main reason for de-assigning domains was that the meaning unit was not related to the target behavior [39]. Finally, to ensure transparency, we calculated agreement between the coders’ initial allocation of meaning units for the first 3 interviews (J.W.K. and M.M.P.) and the last four interviews (M.M.P. and R.B.). This was performed by simple percentage agreement [47]. 

As stated, the aim of the analysis was to identify the barriers and facilitators that were most common. Therefore, inspired by Hsieh and Shannon [44], we considered the frequency of meaning units assigned to a domain as a measure of its importance at the time of the interviews. A domain containing at least 5 percent of the total meaning units was considered important. Finally, these high importance domains were included in an inductive analysis. One researcher (M.M.P.) reviewed the meaning units within each domain to generate collections of meaning units with similar underlying beliefs [39,46]. For each collection of meaning units, a belief statement was generated. A belief statement was defined as a statement with a specific underlying belief [39]. Thus, within each of the theoretical domains all meaning units were grouped with meanings units reflecting the same underlying belief [39]. Additionally, opposing meaning units that were considered to reflect opposing views concerning the same overarching belief were grouped together. 

For example, under the “Beliefs about consequences” domain the responses “I think it’ll be the effect of a physician mentioning it”, and “Well, I have to say, if I talk about it, being a physician, then I think that for a busy nurse it will be prioritized a little more. Or maybe it’ll put a focus on remembering mobility” were grouped under the belief statement “Patients and staff will focus more on mobility if the physician mentions mobility”. The frequency of occurrence of meanings units within each belief statement across all interviews was counted. Another researcher (J.W.K.) subsequently reviewed the belief statements to confirm these.

### 2.6. Ethics

Prior to each interview, we provided information about the background, aim and methods of the study. Oral and written informed consent was obtained from all physicians according to the Helsinki Declaration. We ascertained full anonymity throughout the whole analysis process by allocating an ID-number to each physician instead of using their names. The physicians’ names were only accessible to the members of the research group. According to Danish law, ethical approval is not mandatory for studies not involving biomedical issues and was, therefore, not obtained. However, each interview was conducted by taking the situational ethics into account—e.g., by considering the context’s specific circumstances [48]. Thus, before conducting each interview we would evaluate whether the timing of the interview was bad for the involved physician—i.e., if the physician was too busy and the interview should be rescheduled. 

None of the research members had a working relationship with any of the participating physicians. The study was approved by the Danish Data Protection Agency (AHH-2016-080, I-Suite no.: 05078).

## 3. Results

### 3.1. Informants

Twenty physicians were invited to be interviewed and twelve physicians accepted the invitation (5 male and 7 female). Six of the physicians were employed at X and 6 at Y. The physicians’ working experience ranged from little experience to extensive experience. Their years of experience were as follows: (1) <5 years (*n* = 2); (2) 5–10 years (*n* = 2); (3) 10–20 years (*n* = 2); and (4) +20 years (*n* = 6).

### 3.2. Intercoder Agreement

In the first 3 interviews, 126 meaning units were allocated to one of the 14 domains. Initial agreement was found on 63% of the codes by the two coders, J.W.K. and M.M.P. After consensus discussion, agreement was obtained for 83% of the codes, which were kept for further analysis, and agreement on removal of codes was obtained for 17% of the codes, which were only identified by one coder and which were considered not to be related to the target behavior (Table 1). In the last 4 interviews, 160 meaning units were assigned to one of the 14 domains. Initial agreement on coding was found on 37% of the codes by the two coders, M.M.P. and R.B. After consensus discussion, agreement on coding was obtained for 79% of the codes, and agreement on removal of codes was obtained for 21% of the codes (Table 1).

### 3.3. Key Domains

After the consensus discussions on the meaning units abstracted from the 12 interviews, consensus was obtained for a total of 411 meaning units covering the 14 domains of the TDF. All 14 domains of the TDF were represented in the physicians’ responses regarding mobility. When inspecting the number of responses within each domain, however, the domains Knowledge, Social/Professional role and Identity, Beliefs about consequences, Optimism and Environmental context and resources each contained at least 5 percent of the total responses and were, therefore, as mentioned, defined as domains of high importance (Table 2). In total, these domains covered 70% of the meaning units (*n* = 290).

The meaning units were grouped into 26 belief statements with similar underlying beliefs (Knowledge = 3; Social/Professional role and identity = 4; Optimism = 5; Beliefs about consequences = 7; Environmental context and resources = 7). All belief statements and examples of meaning units for the 5 high importance domains are presented in Table 3.

For example, the following statements were coded under the domain “Beliefs about consequences”: 

“Cross-disciplinary co-operation will be strengthened for sure, because when we prescribe… focus will be put on mobility and we will discuss it. no doubt, it will strengthen our focus and cross-disciplinary co-operation”.(Physician no.3, Department X)

“It’s not at all unlikely that it (red.: Prescription of WALK-plans) will lead to more communication between physicians and physiotherapists. If one has prescribed intensive mobility and the physiotherapist thinks that the patient is unable then, of course, there’ll be communication about it”. (Physician no. 4, Department X)

Both statements were considered to reflect the same overarching belief that the intervention would induce improved cross-professional cooperation. Additionally, the following statements, which were considered to reflect opposing views on the same overarching belief, were grouped together under the “Optimism” domain as a belief statement regarding a belief or disbelief in the effect of the intervention: 

“I have my doubts (red.: about the effect of the intervention). I have my doubts. I have my doubts…”. (Physician no. 1, Department X)

“I have a lot of positive thoughts about it (red.: the intervention). And hmm, I will not doubt your knowledge that a higher degree of mobility during hospitalization probably will expedite discharge and better the prognosis for the patient”. (Physician no. 4, Department X)

### 3.4. Knowledge

The Knowledge domain contained three belief statements: “I know of the intervention and its contents”, “Inactivity can cause functional deficits–mobility is important” and “It’s important to explain to patients what they’re allowed to do and why”. Most of the interviewed physicians had heard about the intervention and were familiar with the contents of the intervention (*n* = 8) (Table 3). Knowledge about the intervention and its contents was obtained and shared at morning meetings among the physicians and at daily conferences, when patient care was discussed.

Furthermore, the majority was aware that inactivity during hospitalization increases the risk of functional decline and the risk of hospital-acquired side effects, and that mobility is important to counteract these possible negative effects (*n* = 8). Some also underlined the necessity of explaining the importance of mobility to patients and relatives, as well as clarifying for patients what they can do while being hospitalized (*n* = 3). Thus, the physicians were knowledgeable about the intervention and its purpose and found instructions to and knowledge sharing with patients necessary.

### 3.5. Social/Professional Role and Identity

As regards the Social/Professional Role and Identity domain, several belief statements evolved: “Bed rest is part of a cultural understanding of what comes with being hospitalized, but patients respect the physician’s words”, “Physicians do not focus on mobility and it’s not their task”, “The communication between physicians and physiotherapists is non-verbal” and “The physiotherapists should assess mobility potential, but the nurses are the main staff in mobilizing the patients”. Firstly, one third of the physicians was confident that patients respect and follow physicians’ recommendations and would be active if a physician told them to be (*n* = 4), although bed rest for some patients and relatives seems to be a part of a cultural understanding of what comes with being hospitalized, and some patients, therefore, consider their bed a “safe nest”.

The physicians were aware that getting the patients out of bed was difficult due to both physical and cultural challenges, and they were confident that the chances of getting the patients up and about were better if encouraged by physicians, as prescribed by the intervention. However, in practice, they did not encourage the patients and did not find this a part of their tasks; they considered this to be a task for other health care professionals. Half of the physicians expressed that their focus as a group was not on mobility and getting the patients out of bed, but on medications, diagnostics, getting the patients well and on discharge (*n* = 6), and some physicians (*n* = 5) clearly expressed that focusing on functioning, mobility and getting patients out of bed was not a job to be undertaken by physicians (*n* = 5). One informant said:

“It’s not part of our medical training (red.: to focus on mobility and functioning). If they’re admitted with pneumonia or diabetes, then that’s what we’ll concentrate on”. (Physician no. 8, Department Y)

However, a few felt that the physicians could play a role in motivating patients (*n* = 2):

“Well, the practical part of it, I do not think we, as physicians, think it’s part of what we do, but we can motivate the patient to get out of bed and say ‘You’re allowed to do that’ or ‘You have to do that because it’s important that you do not lie in bed all the time’”. (Physician no. 2, Department Y)

To the physicians, their job was to prescribe mobility and let physiotherapists and nurses follow up (*n* = 4). Cooperation between nurses and physicians was considered well-functioning, whereas cooperation with the physiotherapists was mostly on a more non-verbal level, through journal notes and prescriptions (*n* = 5). Additionally, there was a clear perception of the necessity of physiotherapists, as opposed to nurses and physicians, to evaluate the patients’ mobility status. Partly because physicians only see snapshots of the patients’ days, whereas the physiotherapists see them daily. However, the nurses were considered the main staff in getting the patients out of bed (*n* = 8). One informant expressed:

“Well, I think the physiotherapists are more important here, because with the clientele we have, mobilizing patients is not always that simple. I mean, most often they’ll need a walking aid, right?”. (Physician no. 1, Department X)

Thus, the physicians had a clear view on the different roles of different health professionals. They expressed that one challenge in increasing mobility might be a cultural understanding that bed rest is part of being hospitalized. They were confident, however, that patients respect and abide by the physician’s words. They also expressed that physicians do not consider patient mobility one of their tasks, and should not focus on functioning, mobility and getting patients out of bed—this should be undertaken by physiotherapists and nurses following physician prescriptions.

### 3.6. Beliefs about Consequences

The Beliefs about Consequences domain revealed the following belief statements: “The intervention will cause improved cross-disciplinary cooperation”, “Patients and staff will focus more on mobility if the physician mentions mobility”, “The intervention will have/will not have positive effects”, “Prescription of WALK-plans will give the physicians more work to do”, and “The physicians will not do it by themselves” (Table 3). The physicians saw the intervention as a means of improving collaboration and communication between physicians, nurses and physiotherapists (*n* = 6), leading to faster and better evaluation of the patient’s abilities to the benefit of the patient. Additionally, some expressed that both patients and staff would be more inclined to focus on mobility if the physician mentioned the importance of mobility (*n* = 4), whereas one physician believed that the effect would be equally as good if presented by a nurse or a physiotherapist. 

The physicians expressed conflicting beliefs about the effect of the intervention. On one hand, they found that the intervention would enable patients to get out of their beds at an earlier stage and recover faster (*n* = 5), and on the other hand, they found that only a minority of their patients would be well enough to comply with the intervention and some would be confused by too much activity in the hall ways (*n* = 4). Furthermore, one third of the physicians expressed concerns about the extra workload in prescribing WALK-plans and believed that they would only do their part of the job if either a nurse or a physiotherapist reminded them to do so (*n* = 4). 

Thus, according to the physicians, the consequences of the intervention would be improved cross-disciplinary cooperation. They were confident that both patients and staff would focus more on mobility if mentioned by the physicians but were not sure the physicians would prescribe mobility since this would be an extra workload. Additionally, some believed in a positive effect of the intervention, whereas others doubted about the effect.

### 3.7. Optimism

In the Optimism domain the following six belief statements evolved: “The intervention is relevant and important”, “I believe/I doubt that the intervention will be effectful/have an effect on mobility”, “The patients are too ill/unable to comply with the intervention”, “It is unrealistic to think that the physicians will prescribe / Prescription is a good idea”, “It’s positive to plan cross-disciplinarily” and “It’s a good idea to use colors on the WALK-plans / The WALK-path is visually appealing” (Table 3). Three of the physicians found the intervention important and relevant (*n* = 3). They expressed that movement is important and that the intervention would be able to put a focus on mobility. Nevertheless, they doubted that enough resources were available. Furthermore, half of the physicians expressed that the intervention would result in enhanced mobility and prevent loss of function and readmissions (*n* = 6). However, two of these physicians expressed doubts about an effect on mobility and an effect of prescribing WALK-plans (*n* = 2). Likewise, one physician found it unrealistic to expect relatives to help (*n* = 1), and another expressed that self-service on clothes would be an infectious time bomb (*n* = 1). Additionally, the physicians doubted that patients in their department, who were believed to be more and more ill, would be able to carry out the intervention (*n* = 5). Several physicians were opposed to having the responsibility for prescribing WALK-plans due to lack of time, an opposition to adding more to the physicians’ work-load, prescribing something evident and adding more things to register in the electronic patient journals (*n* = 6) and found it unrealistic to think the physicians would remember the prescriptions:

“Prescription of mobility will be neglected very quickly—first comes medications, blood samples, food etc. So, I do not think a prescription is enough – I think it’ll be thrown into the corner”. (Physician no. 10, Department Y)

On the other hand, a few said it was worth the try and a good idea (*n* = 4). Furthermore, one third of the physicians found cross-disciplinary evaluation of patients with regards to WALK-plans to be good for cooperation between nurses, physicians and physiotherapists (*n* = 4).

Additionally, some expressed that using levels on the WALK-plan was helpful and that colors and visual cues on the WALK-path and the WALK-plans were visually appealing. One informant expressed it this way:

“It’s easier than something on an individual level, which has to be titrated correctly. This is just a small, medium or large package”. (Physician no. 12, Department Y)

The physicians were optimistic about the relevance of the intervention, but were uncertain about the effect on mobility, since they found the patients too ill, and were uncertain about whether they would take time to prescribe mobility. They were also optimistic about cross-disciplinary planning and about using colors to indicate levels on WALK-plans.

### 3.8. Environmental Context and Resources

As regards the Environmental Context and Resources domain the belief statements were: “Due to lack of time, mobility will be forgotten in the bustle”, “The interior of the department does not fit with mobility”, “The patients are too ill”, “The physicians only see the patients very briefly and need the physiotherapists to evaluate function”, “We only get a snapshot of the patients because they have to be discharged fast”, “Too much time is used in front of the computer”, and “Self-service on clothes can spare time but is against regulations and a contagious bomb” (Table 3). A general belief among the physicians was that lack of time, excessive work load and lack of resources/staff would hinder a focus on mobility (*n* = 10), which would be forgotten in the everyday bustle and downgraded compared to other tasks—i.e., dispensing medicine (*n* = 7). However, some stated that it should be feasible and just be a matter of taking the time (*n* = 4):

“Most physicians are busy doing rounds on a lot of patients but telling a patient that it would be beneficial to get up and walk a bit, I mean, if that’s all you have to say and not much more, it only takes 20 s. So, it should be possible to take out time for that, I guess”. (Physician no. 6, Department X)

Most of the physicians believed that the department’s interior did not fit with a wish to enhance mobility due to uninspiring and faulty décor, and lack of space, which would make the patients insecure in the hallway (*n* = 10). One expressed it this way:

“If you walk badly and are a little dizzy and so on, it’s not interesting to walk in a hallway which is narrow and where people come rushing. Then I think you prefer staying in bed. There you’ll feel safe”. (Physician no. 11, Department Y)

A few, on the contrary, expressed that there was room for walking in the hallway and a WALK-path would be used (*n* = 2). In addition, some of the physicians found the patients admitted to the department were too old and ill to focus on mobility (*n* = 3). 

Additionally, some physicians expressed that they only see the patients very briefly due to how they are organized (consulting patients in more departments). Therefore, they depend on an assessment by a physiotherapist prior to mobilization (*n* = 4), partly because the physiotherapists should hand out walking aids, and partly because the physiotherapists know more about the patients’ functioning and have more competencies in evaluating function than the physicians, who only come by on supervision (*n* = 4).

The fact that the patients are discharged quickly, partly for economic reasons, was a factor believed to counteract a focus on mobility (*n* = 3), along with an excessive amount of time spent in front of the computers (*n* = 4). Thus, the physicians agreed that they lacked both time and resources, and spent too much time in front of computers, which is why they would forget about mobility. They expressed a need for physiotherapists to evaluate the patients’ function, since they only saw the patients very briefly themselves. They also found the department unfit for mobility, but believed that self-service could spare time.

### 3.9. Barriers and Facilitators

Several different barriers and facilitators emerged. With regard to barriers, the physicians believed that only a minority of their patients would be able to comply with the intervention (Optimism), and they doubted that the department’s interior was suitable for promoting mobility (Environmental context and resources). They expressed conflicting thoughts about the likelihood that their patients would be able to follow the intervention and that sufficient resources were available (Beliefs about consequences; Optimism). Despite a belief in a physician effect, their focus was on other tasks—i.e., medication, diagnostics and discharge (Social/Professional Role and Identity), and they believed that lack of time and resources would hinder a physician focus on mobility (Environmental context and resources; Optimism). Several physicians were also opposed to putting more workload on the physicians’ shoulders and doubted that they would remember to do their part of the job unless reminded to do so (Social/Professional Role and Identity; Beliefs about consequences). They expressed mixed feelings about the responsibility for promoting mobility and did not consider functioning and mobility as part of their job (Social/Professional Role and Identity), which they expected other health care professionals to handle (Social/Professional role and identity). Particularly the physiotherapists were considered to be responsible for assessing mobility status and evaluating what could be expected of each patient, whereas nurses were considered the main staff in securing mobility, but the physicians mainly communicated with the physiotherapist at a non-verbal level (Social/Professional role and identity; Environmental context and resources). 

Concerning facilitators, most of the physicians expressed an awareness of the negative consequences of inactivity during hospitalization (Knowledge), and they believed that the intervention could promote faster recovery (Beliefs about consequences). They expressed that enhanced focus on mobility was important, a good idea and would prevent loss of functioning and readmission of their patients (Knowledge). They also saw the intervention as a means of improving collaboration and communication between physicians, nurses and physiotherapists (Beliefs about consequences, Optimism), and they were confident that the patients would get out of bed if the physician told them to. The same response would be elicited by nurses and physiotherapists who would focus more on mobility if the physician asked them to do so (Social/Professional Beliefs about consequences). 

## 4. Discussion

In this study, we used the TDF [39] to explore barriers and facilitators to physicians’ promotion of mobility in the WALK-Cph mobility intervention. The results show that the identified barriers and facilitators were categorized primarily in five of the 14 domains of TDF: Knowledge; Social/Professional role and identity; Optimism; Beliefs about consequences and Environmental context and resources domains. These domains reflect both individual and more social influences on physicians’ attitudes and behaviors to achieve increased mobility in hospitalized older medical patients. Thus, whereas Knowledge, Optimism and Beliefs about consequences are associated with physicians’ individual cognitions, Social/Professional role and identity and Environmental context and resources are more related to external influences on physicians’ attitudes and behaviors—e.g., expectations on their behaviors in various work situations and circumstances in their work environment that discourage or encourage certain behaviors. This finding suggests the importance of targeting both internal (i.e., individual) and external levels to influence physicians’ attitudes and behaviors of relevance for the promotion of mobility. 

In the present study, the physicians were aware of the necessity and relevance of focusing on in-hospital mobility. Despite a belief in a “physician effect”, they thought that nurses should be the main staff responsible for mobility after physiotherapist assessment of functioning. In line with this, nurses have previously been identified as the key health care professionals in promoting mobility in hospitalized patients as they are the health care professionals spending most time with the patients [49].Additionally, the perception that mobility, as part of patient care, must be undertaken by nurses is well in line with the professional roles historically undertaken by nurses and physicians, where nurses focus on the health and wellbeing of the individual patient, whereas physicians focus on diagnoses and ameliorating medical conditions [50]. This cultural and social understanding of the roles of different health care professionals could hinder patient mobility since physicians, as we found in this study, may expect nurses and therapists to undertake tasks concerning mobility, whereas nurses may not consider mobility their responsibility [19,51] and, therefore, infrequently initiate mobility [52] or refrain from doing so due to e.g., fear of patient falls [33]. Thus, mobility ends up between two stools.

The physicians in the present study stated that patients respect the physicians’ words and would move about if told to do so by a physician. This is confirmed by studies reporting that older adults find it important that a physician tells them to prioritize mobility [32,53,54,55,56,57,58]. However, in line with this study, previous studies have reported that physicians seem to refrain from advising older adults to be physically active as part of their treatment [28,32,59]. In an ethnographic field study conducted as a baseline study for the WALK-Cph project, Kirk et al. [19] found that different health care professionals attach different meanings to the concept of mobility. It was found that mobility of older medical patients is entangled in a network of different professional identities, which blur the responsibility for mobility. Although considered an important part of treatment, nurses, nursing assistants and physicians did not consider mobility as part of their core tasks [19]. Additionally, as opposed to physicians educated in geriatric specialties, who focused on mobility, physicians educated in medical specialties primarily focused on diagnosis, treatment and discharge of the patient, indicating that their medical education influences their view on what is considered part of their core tasks [19,60]. Furthermore, an Irish study investigating physician attitudes towards mobility counselling found an association between considering activity counselling your role and providing this counselling [60], and revealed a lack of physician training in mobility. Both factors may apply to the physicians in the present study. 

Additionally, as stated by Casanova et al., the professional cultures of nurses and physicians have led to different role perceptions and an imbalance in authority, where physicians assume a directive role founded on the science of medicine, whereas nurses take on a more accommodating role with a focus on care of the individual patient—disparate roles that present challenges to effective cross-professional collaboration [61]. However, the physicians in the present study believed that the intervention could promote cross-disciplinary collaboration between nurses, physiotherapists and physicians. This finding is in line with a study by Vazirani et al. [62] showing that a multidisciplinary intervention in an acute medical unit resulted in better communication and collaboration between nurses and physicians, and therefore, a focus on bettering cross-professional collaboration may indirectly enhance patient mobility. 

The physicians mentioned several factors that they considered barriers for patient mobility—e.g., a cultural understanding of bedrest as a part of being hospitalized, the patients’ illness, the hospital environment, lack of time and resources and professional roles (i.e., physicians have other tasks to focus on). These barriers are well in line with previous studies investigating health care professionals’ attitudes and behaviors concerning in-hospital mobility in older medical patients. Barriers for patient ambulation have been reported to be lack of time and staffing [19,31,33,63], severity of patient illness and symptoms [31,33], fear of patients falling [33,63], the hospital environment [31,33] and professional roles [19,31,63]. Some of these factors have been reported by patients as well. Studies have shown that while older patients find it appropriate to exercise, they do not expect to be doing so during hospitalization [32]. This is mainly due to barriers such as illness symptoms, fear of falling, discomfort of medical devices and a bed centered environment [31,32,33]. 

Additionally, the physicians stated that much time was spent on administrative tasks in front of computers at the expense of other tasks. According to Danish law, physicians are required to register all treatment and contacts in the patient’s electronic journal. In 2017, a new Health care Platform for all registrations was implemented, and therefore, extensive changes were made to the workflow regarding journaling—i.e., the physicians should now register all their actions directly in the electronic journal instead of asking a secretary to transcribe what had been dictated. This change has led to physicians spending more time on journaling than previously. 

### Strengths and Limitations

This study has several strengths worth mentioning. Firstly, the study uses a validated theoretical framework as the basis of investigating physicians’ barriers and facilitators for promoting mobility in older adults. This framework provides a satisfactory theoretical coverage of potential implementation determinants—i.e., barriers and facilitators to successful implementation of practice changes, therefore providing a relevant basis for developing the interview guide [39]. Secondly, the informants in the present study have a wide range of working experience, likely contributing to a broad range of views on mobility. Thirdly, to avoid researcher bias in the data collection and data analysis and assure credibility, we used researcher triangulation where three independent researchers coded the data and discussed the coding until consensus was reached [64,65].

The study has several limitations, which should be considered when interpreting the findings. Firstly, we saw a low initial agreement between coders (58% and 24%, respectively), which can be a sign of complex statements [37]. After consensus discussion, however, negotiated agreement was obtained for 83 % and 67 % of the codes, respectively. These intercoder agreements are in line with previously reported initial and negotiated percentage agreements [66,67]. Additionally, the lack of full negotiated agreement is merely a sign of de-assignment of codes not related to the target behavior. Secondly, we interviewed all available physicians in the involved departments, but it is unknown whether we interviewed enough physicians to obtain data saturation. However, Guest et al. (2006) found that an average of 12 interviews is usually sufficient to obtain data saturation [68]. Thirdly, we chose to focus on the determinant domains covering at least 5 % of the meaning units. However, it is possible that the most frequently identified barriers and facilitators may not be the barriers and facilitators of highest importance. Thus, there is a risk that some significant barriers and facilitators were omitted [69]. Additionally, the extensive interview guide may have made it difficult to obtain in-depth information since the guide may steer the interviewer away to new questions instead of allowing enough time for the informant to really explore each question. Furthermore, it is likely that providing information about the contents of the intervention before carrying out the interviews (concerning physicians’ promotion of mobility) may have influenced the physicians’ responses towards a more positive attitude regarding physician-promoted mobility. Moreover, using qualitative content analysis with a quantitative element, thereby focusing on manifest content, implies a risk of losing in-depth information [42,43].

## 5. Conclusions

In conclusion, we used the Theoretical Domains Framework to investigate the barriers and facilitators of physicians concerning mobility in hospitalized older medical patients. The physicians expressed barriers for mobility in older medical patients to be patients who are too ill, an unfit department, a culture of bedrest, professional roles and responsibility (i.e., mobility is not part of the physicians’ job and responsibility), lack of time and resources and unwillingness to accept an extra workload. The facilitators for mobility mentioned by the physicians were enhanced cross-professional cooperation focusing on mobility, physician encouragement of mobility towards other health care professionals as well as patients (i.e., a physician effect) and patient self-service to relieve the health care professionals. These barriers reflect different system levels of influences on physicians’ behaviors, from the individual physician (optimism, beliefs about consequences) and the profession (social/professional role and identity) to the broader context in which the physicians work (environmental context and resources). Thus, it is recommended that future interventions aiming to increase mobility in hospitalized older medical patients should target multiple levels to influence physicians’ promotion of mobility.

## Figures and Tables

**Table 1 geriatrics-05-00074-t001:** Intercoder agreement on meaning units.

TDF Domain	A: Meaning Units (no.)	A: Initial Consensus (no. (%))	A: Negotiated Consensus (no.)	B: Meaning Units (no.)	B: Initial Consensus (no. (%))	B: Negotiated Consensus (no.)
Knowledge	8	4 (50)	7 (87.6)	9	5 (55.5)	8 (88.9)
Skills	8	2 (25)	6 (75)	2	0 (0)	0 (0)
Social/professional role and identity	12	10 (83.3)	11 (91.7)	36	18 (50)	33 (91.7)
Beliefs about capabilities	8	6 (25)	7 (87.5)	5	1 (20)	2 (40)
Beliefs about consequences	9	7 (77.8)	8 (88.9)	11	0 (0)	9 (81.8)
Optimism	14	7 (50)	12 (85.7)	18	3 (25)	15 (83.3)
Reinforcement	5	3 (60)	4 (80)	4	1 (25)	3 (75)
Intentions	4	2 (50)	4 (50)	5	0 (0)	1 (20)
Goals	8	5 (62.5)	6 (75)	8	1 (12.5)	8 (100)
Memory, attention and decision processes	3	2 (66.7)	2 (66.7)	12	5 (41.7)	9 (75)
Environmental context and resources	27	20 (74.1)	22 (81.5)	39	22 (56.4)	32 (82.1)
Social influences	4	3 (75)	3 (75)	3	1 (33.3)	1 (33.3)
Emotion	10	5 (50)	8 (80)	1	0 (0)	1 (100)
Behavioral regulation	6	4 (66.7)	5 (83.3)	7	2 (28.6)	5 (71.4)
Total (no. (mean %))	126	80 (63.5)	105 (83.3)	160	59 (36.9)	127 (79.4)

Abbreviations: TDF = Theoretical Domains Framework. Note: This table demonstrates the intercoding agreement between researchers (M.M.P. and J.W.K.) for interviews no. 1–3 (A) and between researchers (M.M.P. and R.B.) for interviews no. 9–12 (B).

**Table 2 geriatrics-05-00074-t002:** Distribution of meaning units across Theoretical Domains Framework domains.

Domain	No. of Meaning Units	Percentage
*Knowledge*	*25*	*6.08%*
Skills	12	2.92%
*Social/Professional role and identity*	*70*	*17.03%*
Beliefs about capabilities	12	2.92%
*Optimism*	*56*	*13.63%*
*Beliefs about consequences*	*39*	*9.49%*
Reinforcement	11	2.68%
Intentions	6	1.46%
Goals	19	4.62%
Memory, attention and decision processes	13	3.16%
*Environmental context and resources*	*100*	*24.33%*
Social influences	17	4.14%
Emotion	13	3.16%
Behavioral regulation	18	4.38%
Total	411	100%

Note: This table demonstrates the distribution of meaning units across Theoretical Domains Framework domains. Domains covering 5% or more of the meaning units are written in italic.

**Table 3 geriatrics-05-00074-t003:** Belief statements within the 5 domains identified as being of high importance (i.e., reflecting 5% or more of the meaning units) and examples of meaning units.

Domain	Belief Statement	Examples of Meaning Units	No. of Meaning Units	No. of Physicians Expressing the Belief Statement
Knowledge	I know of the intervention and its contents	“Yes, I’ve heard about the intervention… Yes, I believe I know about its contents” (Physician no. 2, Department X) “I’ve heard about it at our morning conferences when one of your colleagues told us about it… so, I know about its contents, roughly” (Physician no. 6, Department X)	10	8
	Inactivity can cause functional deficits – mobility is important	“But I think we all think it’s really really important (red. that older adults move) – I’m pretty confident that we all know it is essential” (Physician no. 2, Department X) ”They lose their functioning when they lie in bed and the recovery time is very long afterwards, which increases the risk of pneumonia, UTI’s and DVT’s” (Physician no. 9, Department Y)	12	8
	It’s important to explain to patients what they’re allowed to do and why	“Some don’t know what they’re allowed to and what they’re not allowed to” (Physician no. 3, Department X)	3	3
Social/Professional role and Identity	Bed rest is part of a cultural understanding of what comes with being hospitalized, but patients respect the physician’s words	“It’s one of the challenges in this project. There’s a cultural understanding that being ill means staying in bed” (Physician no. 3, Department Y) “Yes, I think it’s important that the physician brings the message. I’ve experienced that if the nurse tells them to, not much happens, being they’re so authoritarian” (Physician no. 10, Department Y)	7	4
	The communication between physicians and physiotherapists is non-verbal	“Not very much (red.: do I talk with the physiotherapists). Honestly, I don’t really know what happens after I ask for an evaluation by a physiotherapist. I do rounds with the nurses… and we agree on who will profit from rehabilitation to reach their pre-hospitalization level… And then I assume that everything is ok” (Physician no. 5, Department X) “We actually do not communicate that much with the physiotherapists. Sometimes, we prescribe physiotherapy or occupational therapy” (Physician no. 4, Department X)	10	5
	The physiotherapists should assess mobility potential, but the nurses are the main staff in mobilizing the patients	“In my opinion, the physiotherapists are best at evaluation how much they (red.: the patients) can train here and at home” (Physician no. 5, Department X) “… and I think the nurses are the main group (red.: in mobilizing the patients) and in … helping the patients in mobilizing themselves (Physician no. 7, Department Y)	15	8
	Physicians do not focus on mobility and it’s not their task	“We forget about it. Do not focus on it. And we probably do not consider it physician work because it has nothing to do will illness but with maintaining functioning” (Physician no. 9, Department Y) “Well, my focus will be to get a physiotherapist to evaluate the patient and then she’ll take it from there and figure out what needs to be done. So, what I do is to prescribe physiotherapy (Physician no. 11, Department Y)	38	10
Beliefs about consequences	The intervention will cause improved cross-disciplinary cooperation	“Cross-disciplinary co-operation will be strengthened for sure because when we prescribe… focus will be put on mobility and we will discuss it. No doubt, it will strengthen our focus and cross-disciplinary co-operation (Physician no. 3, Department X) “It’s not at all unlikely that it (prescription of walk-plans) will lead to more communication between physicians and physiotherapists. If one has prescribed intensive mobility and the physiotherapist thinks the patient is unable then there’ll of course be communication about it” (Physician no. 4, Department X)	11	7
	Patients and staff will focus more on mobility if the physician mentions mobility	“I think it’ll be the effect of a physician mentioning it (red.: that the patient should get out of bed)… but it’s kind of a paradox… that the nurse tells the physician ’remember to tell the patient to get out of bed, ‘cause when I tell them to they don’t want to’” (Physician no.1, Department X) “Well, I have to say, if I talk about it, being a physician, then I think that for a busy nurse it will be prioritized a little more. Or maybe it’ll put a focus on remembering mobility” (Physician no.9, Department Y)	8	5
	The intervention will have/will not have positive effects	“I think it (red.: a mobility intervention) can provide many positive effects – not just during hospitalization, but also after” (Physician no. 4, Department X) “But as we talked about at our morning conference, many of our patients are so ill that walking is not realistic” (Physician no. 2, Department X)	16 (6/10)*	10 (5/5)*
	Prescription of walk plans will give the physicians more work to do	“The consequence can be that we need more clicks (red.: with the computer mouse). So, it’ll be more work” (Physician no. 4, Department X)	2	2
	The physicians will not do it by themselves	“… it has to be a nurse. I’m not sure the physicians will stick with it” (Physician no. 8, Department Y)	2	2
Optimism	The intervention is relevant and important	”I think the intervention is extremely relevant… we have to start somewhere, so I think it could light a small candle of awareness so that people could start to think about it (red.: moving) and start doing something” (Physician no. 6, Department X) ”… when you hear about this you cannot help but wonder if we have enough resources. I’m maybe a little sceptic about that aspect, but I definitely advocate for prioritizing this – also with resources” (Physician no. 4, Department X)	7	3
	I believe/I doubt that the intervention will have an effect on mobility	“I do not doubt that increased focus on mobility during hospitalization will accelerate discharge and better the patient’s prognosis” (Physician no. 4, Department X) “I have my doubts (red.: about the effect of an intervention on mobility). I have my doubts. I have my doubts” (Physician no.1, Department X)	13 (6/7)*	8 (6/4)*
	The patients are too ill/unable to comply with the intervention	“One forgets that the populations is getting older and older and those who are in our beds are more and more ill” (Physician no.1, Department X)	7	5
	It’s unrealistic to think that the physicians will prescribe / Prescription is a good idea	”Well, I’ll be happy to prescribe” (Physician no. 5, Department X) “A thing like a prescription on mobility, it’ll quickly be neglected in favor of medication and blood samples and food…” (Physician no. 10, Department Y)	17 (12/6)*	11 (6/5)*
	It’s positive to plan cross-disciplinarily	“For the cross-disciplinary cooperation it’s the same. After all, you can make a better plan together if you agree on what the patient is able to do. So, if this is feasible then I only consider it as something positive” (Physician no. 5, Department X)	5	3
	It’s a good idea to use colors on the walk-plans / The walk-path is visually appealing	“It’s easier if it’s just a big, medium and small package” (Physician no. 12, Department Y) ”I think it’s a really good idea that there’s something on the path and on the posters, arrows, feet… it lures you to get started” (no. 12, Department Y)	5	5
	Other	“We’re in opposition to desk theory” (Physician no. 1, Department X)	2	2
Environmental context and resources	Due to lack of time mobility will be forgotten in the bustle	“And primarily it’s the busy working day which could result in one skipping precisely that (red.: mobility) when there are a thousand other things one has to do, which all only take half a minute” (Physician no. 2, Department X) “If they need help to get up, the nurses won’t have the resources – they’re busy as flies… no busy as bees” (Physician no. 4, Department X)	45	12
	The interior of the department does not fit with mobility	”Well, it’s nice to lie in bed and watch television. The patient rooms have really modern TVs. And all patients have their own and can watch all sorts of things. Everybody has iPads. So, it doesn’t exactly encourage you to get up and move” (Physician no. 6, Department X) ”… if you walk badly and are a little dizzy and so on, it’s not interesting to walk in a hallway which is narrow and where people come rushing. Then I think you prefer staying in bed. There you’ll feel safe” (Physician no. 11, Department Y)	27	9
	The patients are too ill	“You tend to forget that the patient population gets older and older and they are more and more ill, those who are in our beds” (Physician no. 1, Department X) “… those who’re admitted are at least 80 years old and not able to stand on their legs…” (Physician no. 7, Department Y)	9	3
	The physicians only see the patients very briefly and need the physiotherapists to evaluate function	”With the patients we have we often ask for a physiotherapist to evaluate the need for a walking aid. And it’s maybe a little foolish that we have to ask for an evaluation before starting mobilization (Physician no. 1, Department X) “It has to be based on physiotherapist evaluation. I would not… eh… have the time… and I would need to know what is realistic before I ask the patient to walk” (Physician no. 11, Department Y)	4	4
	We only get a snapshot of the patients ‘cause they have to be discharged fast	“I’m asked to discharge him the moment he can get out of bed… it’s expensive beds…” (Physician no. 7, Department Y) ”You get these little snapshots. You rarely follow the patients. They come in quickly and leave quickly and… eh… you have changing tasks as a physician, which means that you’re in a lot of different places and cannot follow the patient, really, in relation to such a plan. It’s difficult” (Physician no. 11, Department Y)	8	3
	Too much time is used in front of the computer	”Rounds get to be about the electronic patient journal and not an evaluation of the patient. Often rounds take place in the computer […] often the patient doesn’t notice that we’ve been by on rounds because everything has to be so fast. Back in the days, we worked in soap bubbles. When we had rounds, we were in a bubble and only something really acute could disturb us. Focus was on the patient and the situation” (Physician no. 3, Department X) ”Another limiting factor is that, well, a lot do rounds by sitting by the computer and make notes on a patient list […] and use it as a working paper during rounds with the nurses […] and the you have to go back to the computer to register and finish the round afterwards […]” (Physician no. 4, Department X)	5	4
	Self-service on clothes can spare time but is against regulations and a contagious bomb	”[…] we have had big problems with resistant bacteria […]. So asking patients to pick up clothes would potentially be a contagious bomb” (Physician no. 9, Department Y)	3	3

Sample quotes: Examples of quotes considered to reflect the core thought. *numbers in parenthesis reflect opposing views (e.g., 6 physicians express that the intervention will have an effect, and 10 express that the intervention will not have an effect.

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
