# Peer review of "Is Promotion of Mobility in Older Patients Hospitalized for Medical Illness a Physician’s Job?—An Interview Study with Physicians in Denmark"

_geriatrics, 2020, doi:10.3390/geriatrics5040074_

Round 1

Reviewer 1 Report

Thank you for allowing me to read your interesting study. Below are comments that I feel need to be addressed to strengthen your paper:

Study design: it is critical to indicate in this section what type of content analysis was conducted and which content analysis author the research team followed to guide data collection and analysis structure. Based on the description, use of a theoretical framework- the authors used a directed content analysis- but this is not stated

Study design lines 90-93- “One of these components encompassed that the physicians should prescribe WALK-plans to all patients who were capable of walking and they should encourage all patients to be physically active during hospitalization”. I do not see the connection with this content and study design- this section should describe in detail the qualitative approached used and rationale for why that approach was chosen.

Participants: what is the structure of physician work in the hospital- do the physicians interviewed provide primary care and acute care? Or did the physicians interviewed only provide hospital based care?

How many physicians were available to recruit from? What was the total sample size?

Did you exclude physicians during recruitment based on wanting to have a diverse sample? This reviewer is a bit concerned that the head physicians hand selected and encouraged certain physicians to participate. how was this controlled for?

Semi- structured interviews: please provide rationale for providing information on physician promotion of mobility prior to the interviews? Please address if providing this information a priori influenced or framed how your participants responded to questions.

Line 133- did responses address the emotion domain?

Interviews: did all you ask all participants the same questions? Or did questions change between participants?

How many total questions were asked?

Line 137-141- looks like there was 2 purposes for this study- 1) to understand barriers and facilitators and 2) to explore physican’s perceptions of feasibility. Given this what percent of the interviews were for purpose 1 and purpose 2?

Line 158- manifest content has also been described as transcribed text-

Data analysis: given that a deductive content analysis approach was used did you use an unstructured analysis matrix or a structured analysis matrix to aid in the data analysis process?

Line 160: What do you mean by iterative process based on the TDF- please explain?

Line 162- please explain what you mean by process supervisor

Line 168: more information is needed about what the coding strategy was and how it was used

Line 171: please provide a more detailed description of what is a meaning unit and how these align with the TDF. How did the team identify the meaning units? Looks like constant comparative analysis was used to differentiate meaning units- is this what was done? Please provide more detail about your coding strategy- what was it?

Line 176: what was your percent agreement between coders? Did MMP serve as the gold standard or expert that the other coders were compared to?

The whole section that contains information about how meaning units were used is very confusing and hard to follow impacting the reviewer’s ability to understand what was done and if the analysis adhered to rigorous standards. This section needs improvement

Line 183- using the phrase “frequently reported” is too quantitative for a qualitative study. How often you count something does not equate with importance- major flaw that needs to be addressed. I am concerned the team is quantifying qualitative data or forcing quantitative principles into a qualitative study. Being able to generate more codes could simple be due to how the question was asked which could stimulate more discussion. Not generating as many codes could also be a problem with how the question was asked which may have shut down responses from participants.

Line 191: what is meant by “belief statements”? what are these

Line 192-193 are you referring to a core or main category?

Ethics section: did members of the research team have a working relationship with physician participants? If so, this needs to be disclosed. How did the team address confidentiality of the data?

Intercoder agreement: again really confused about what is a meaning unit- is this a label assigned? Also it is always a struggle when quantitative methods are being forced into a qualitative study- as a reviewer telling me % agreement leaves many more questions. It looks like codes were simply tossed out- did the research team return to the data where analysts did not agree and recode together occur? Based on % it appears that 50% of codes were tossed out!

Line 228: prior you identified a total of 286 (126 from 3 interviews and 160 from the last 4 interviews) and in this line you indicate that there were 411 meaning units- which is it? This needs to be clarified

Table 3: there is a difference between awareness (know of the intervention) and knowledge- how are you differentiating between these 2 constructs

Table 3: social/professional role and identity- the section on bedrest- is the participant referring to cultural understanding among physicians that being ill means staying in bed or cultural understanding of patients that being ill means staying in bed- if it is a cultural understanding among patients- what does this have to do with social/professional role identify?

Table 3: belief statement the cooperation between physicians and physio is non-verbal: the quote provided does not speak to cooperation between physicians and physiotherapist- it indicates there is an assumption that work will be done. The second quote also does not support cooperation- the physician disregards the physio contributions!

Results: in this section there is reference to “themes”. There is much variation in how Themes are worded and how they are applied. But in general, and most scholars agree, that Themes should represent the abstract that brings meaning and identity to a participants experience and captures the wholeness of data that falls within the theme. Sometimes the authors state belief statements and other times they state themes within the apriori identified domain- this is very confusing. When theme is used is represents the data superficially or rather presented as a statement- for example line 338-339: “The intervention will cause improved cross-disciplinary cooperation”. This is not stated as an abstraction. I suggest the research go back and review how they are using the concept Theme in the results section.

Discussion: the authors continue to present results in the discussion section lines 479-504.

Study design: it is critical to indicate in this section what type of content analysis was conducted and which content analysis author the research team followed to guide data collection and analysis structure. Based on the description, use of a theoretical framework- the authors used a directed content analysis- but this is not stated

Study design lines 90-93- “One of these components encompassed that the physicians should prescribe WALK-plans to all patients who were capable of walking and they should encourage all patients to be physically active during hospitalization”. I do not see the connection with this content and study design- this section should describe in detail the qualitative approached used and rationale for why that approach was chosen.

Participants: what is the structure of physician work in the hospital- do the physicians interviewed provide primary care and acute care? Or did the physicians interviewed only provide hospital based care?

How many physicians were available to recruit from? What was the total sample size?

Did you exclude physicians during recruitment based on wanting to have a diverse sample? This reviewer is a bit concerned that the head physicians hand selected and encouraged certain physicians to participate. how was this controlled for?

Semi- structured interviews: please provide rationale for providing information on physician promotion of mobility prior to the interviews? Please address if providing this information a priori influenced or framed how your participants responded to questions.

Line 133- did responses address the emotion domain?

Interviews: did all you ask all participants the same questions? Or did questions change between participants?

How many total questions were asked?

Line 137-141- looks like there was 2 purposes for this study- 1) to understand barriers and facilitators and 2) to explore physican’s perceptions of feasibility. Given this what percent of the interviews were for purpose 1 and purpose 2?

Line 158- manifest content has also been described as transcribed text-

Data analysis: given that a deductive content analysis approach was used did you use an unstructured analysis matrix or a structured analysis matrix to aid in the data analysis process?

Line 160: What do you mean by iterative process based on the TDF- please explain?

Line 162- please explain what you mean by process supervisor

Line 168: more information is needed about what the coding strategy was and how it was used

Line 171: please provide a more detailed description of what is a meaning unit and how these align with the TDF. How did the team identify the meaning units? Looks like constant comparative analysis was used to differentiate meaning units- is this what was done? Please provide more detail about your coding strategy- what was it?

Line 176: what was your percent agreement between coders? Did MMP serve as the gold standard or expert that the other coders were compared to?

The whole section that contains information about how meaning units were used is very confusing and hard to follow impacting the reviewer’s ability to understand what was done and if the analysis adhered to rigorous standards. This section needs improvement

Line 183- using the phrase “frequently reported” is too quantitative for a qualitative study. How often you count something does not equate with importance- major flaw that needs to be addressed. I am concerned the team is quantifying qualitative data or forcing quantitative principles into a qualitative study. Being able to generate more codes could simple be due to how the question was asked which could stimulate more discussion. Not generating as many codes could also be a problem with how the question was asked which may have shut down responses from participants.

Line 191: what is meant by “belief statements”? what are these

Line 192-193 are you referring to a core or main category?

Ethics section: did members of the research team have a working relationship with physician participants? If so, this needs to be disclosed. How did the team address confidentiality of the data?

Intercoder agreement: again really confused about what is a meaning unit- is this a label assigned? Also it is always a struggle when quantitative methods are being forced into a qualitative study- as a reviewer telling me % agreement leaves many more questions. It looks like codes were simply tossed out- did the research team return to the data where analysts did not agree and recode together occur? Based on % it appears that 50% of codes were tossed out!

Line 228: prior you identified a total of 286 (126 from 3 interviews and 160 from the last 4 interviews) and in this line you indicate that there were 411 meaning units- which is it? This needs to be clarified

Table 3: there is a difference between awareness (know of the intervention) and knowledge- how are you differentiating between these 2 constructs

Table 3: social/professional role and identity- the section on bedrest- is the participant referring to cultural understanding among physicians that being ill means staying in bed or cultural understanding of patients that being ill means staying in bed- if it is a cultural understanding among patients- what does this have to do with social/professional role identify?

Table 3: belief statement the cooperation between physicians and physio is non-verbal: the quote provided does not speak to cooperation between physicians and physiotherapist- it indicates there is an assumption that work will be done. The second quote also does not support cooperation- the physician disregards the physio contributions!

Results: in this section there is reference to “themes”. There is much variation in how Themes are worded and how they are applied. But in general, and most scholars agree, that Themes should represent the abstract that brings meaning and identity to a participants experience and captures the wholeness of data that falls within the theme. Sometimes the authors state belief statements and other times they state themes within the apriori identified domain- this is very confusing. When theme is used is represents the data superficially or rather presented as a statement- for example line 338-339: “The intervention will cause improved cross-disciplinary cooperation”. This is not stated as an abstraction. I suggest the research go back and review how they are using the concept Theme in the results section.

Discussion: the authors continue to present results in the discussion section lines 479-504.

Author Response

Dear Reviewer 1

Please find our response to your comments below:

Comment 1: 

Study design: it is critical to indicate in this section what type of content analysis was conducted and which content analysis author the research team followed to guide data collection and analysis structure. Based on the description, use of a theoretical framework- the authors used a directed content analysis- but this is not stated

Reply to comment 1:

We thank the reviewer for revealing this lack of clarity in the manuscript. We have rearranged the manuscript according to the SRQR guideline and have clarified the use of a directed content analysis. Therefore, we have added “directed” and a reference for the use of this method (Hsieh H-F and Shannon SE. Three Approaches to Qualitative Content Analysis. Qualitative health research. Vol. 15. No. 9. November 2005; 1277-1288). We did not, however, have a goal of validating or extending the theoretical framework

Action taken:

We have written the following in the beginning of the “Data analysis” paragraph:

We conducted a qualitative directed content analysis with a focus on manifest content [1–3]. According to Joffe and Yardley, manifest content refers to what is directly observable and visible in the data [2]”.

Comment 2:

Study design lines 90-93- “One of these components encompassed that the physicians should prescribe WALK-plans to all patients who were capable of walking and they should encourage all patients to be physically active during hospitalization”. I do not see the connection with this content and study design- this section should describe in detail the qualitative approached used and rationale for why that approach was chosen.

Reply to comment 2:

We agree with the reviewer that the paragraph should not be part of the “Study design” section. We have removed the information from this paragraph and have included this information in the introduction. Also, we have changed the structure of the manuscript to comply with The Standards for Reporting Qualitative Research (SRQR) (O’Brien et al. Standards for Reporting Qualitative Research: A Synthesis of Recommendations. Academic Medicine, Vol. 89, No. 9 / September 2014).

Action taken:

We have included two paragraphs “qualitative approach” and “semi-structured interviews” under “Study design”. Also, we have added this sentence under “data analysis”:

“A directed content analysis was used to identify statements with content within the predefined TDF domains”.

Comment 3:

Participants: what is the structure of physician work in the hospital- do the physicians interviewed provide primary care and acute care? Or did the physicians interviewed only provide hospital-based care?

Reply to comment 3:

All physicians only provide hospital-based care.

Action taken:

We have included a statement about this under “setting”.

Comment 4: 

How many physicians were available to recruit from? What was the total sample size?

Did you exclude physicians during recruitment based on wanting to have a diverse sample? This reviewer is a bit concerned that the head physician hand selected and encouraged certain physicians to participate. how was this controlled for?

Reply to comment 4:

We thank the reviewer for pointing out this lack of clarity in the manuscript. By email, we invited all physicians with responsibility at the two departments (N=20) to participate in the interviews. Also, two representatives from the WALK-Cph research group (JWK and OA) attended a morning meeting at each department to inform the physicians about the WALK-Cph project and about the emails. Also, we asked the head physicians to mention the emails and the interviews at the following morning meetings to assure that physicians, who had not been able to participate when the WALK-Cph representatives were present were also informed about the email invitations. Thus, the head physicians did not hand select any physicians, who participated in the interview. Of the 20 invited physicians, 4 did not respond to the email (and reminder) and 4 informed orally at the information meetings that they did not wish to participate.

Action taken:

To clarify this recruitment strategy, we have added the following:

“Additionally, two representatives from the WALK-Cph research group (JWK and OA) attended a morning meeting at each department to inform the physicians about the WALK-Cph project and about the interviews and e-mail invitations. At the following morning meeting, the head physicians in both departments reminded the physicians to reply to the e-mails, either with a confirmation to participate or a rejection to participate. Sampling ended when all invited physicians had either accepted or declined the invitation”.

Comment 5:

Semi- structured interviews: please provide rationale for providing information on physician promotion of mobility prior to the interviews? Please address if providing this information a priori influenced or framed how your participants responded to questions.

Reply to comment 5:

The WALK-Cph intervention was co-designed in collaboration between researchers and stakeholders (health care professionals, patients, relatives). Unfortunately, only one physician accepted to attend the workshops. Therefore, the design process was lacked knowledge about barriers and facilitators experienced by physicians regarding the WALK-Cph intervention and their own promotion of mobility or lack hereof. We provided information about mobility prior to the interviews to assure that the interviewed physicians were provided with knowledge similar to the knowledge provided to the workshop participants.

We acknowledge that providing information on mobility a priori may have influenced the physicians’ responses and have added a paragraph to the study limitations.

Action taken:

Added to Study Limitations: “Further, it is likely that providing information about the contents of the intervention before carrying out the interviews (concerning physicians’ promotion of mobility) may have influenced the physicians’ responses towards a more positive attitude regarding physician- promoted mobility”.

Comment 6:

Line 133- did responses address the emotion domain?

Reply to comment 6:

Yes, as stated in Table 2, 3% of the responses addresses the emotion domain.

Comment 7:

Interviews: did all you ask all participants the same questions? Or did questions change between participants?

How many total questions were asked?

Reply to comment 7:

Yes, we asked all participants the same questions. However, as stated in the manuscript additional follow-up questions targeting a specific domain were allowed if elaboration of a response was needed.

A total of 29 questions were asked.

Action taken:

We have added to the manuscript that a total of 29 questions were asked and that all informants received all 29 questions.

Comment 8: 

Line 137-141- looks like there was 2 purposes for this study- 1) to understand barriers and facilitators and 2) to explore physicians’ perceptions of feasibility. Given this what percent of the interviews were for purpose 1 and purpose 2?

Reply to comment 8:

Thank you for pointing out this lack of clarity. The study has one aim, to understand barriers and facilitators, according to the physicians, regarding physicians’ promotion of mobility as part of an intervention, and thus the feasibility of the intervention. This has been re-formulated in the manuscript to clarify. We sought the physicians’ perceptions of the feasibility of mobility promotion and our analysis focused on classifying these into different types of barriers and facilitators regarding the feasibility.

Action taken:

We have changed the wording of the aim to: “The aim of this study was to classify the most common barriers and facilitators to physicians’ promotion of mobility in older medical patients as part of an intervention to promote mobility by the use of the Theoretical Domains Framework” 

Comment 9:

Line 158- manifest content has also been described as transcribed text-

Reply to comment 9:

True. In this study, we use the definition by Joffe and Yardley, who defined manifest content as what is directly observable and visible in the data (Joffe, H.; Yardley, L. Content and thematic analysis. In Research methods for clinical and health psychology.; Sage Publications Inc.: London, 2004; pp. 55–68.)

Comment 10:

Data analysis: given that a deductive content analysis approach was used did you use an unstructured analysis matrix or a structured analysis matrix to aid in the data analysis process?

Reply to comment 10:

We used a structured process guided by the pre-existing domains of the TDF.

Action taken:

None

Comment 11

Line 160: What do you mean by iterative process based on the TDF- please explain?

Reply to comment 11:

By an iterative process we mean that we evaluated our coding multiple times. Firstly, we developed a coding approach based on the first two interview transcripts. We used this approach to guide the coding of the remaining interviews. All interviews were coded independently by 2-3 researchers (JWK, MMP, RB) after which the coding was reevaluated in discussions to reach consensus between all three coders. In the consensus discussions, all three coders re-read the meaning unit in question and all coders explained their reason for categorizing the meaning unit under a specific domain. Hereafter, all coders had the possibility of re-evaluating their decision and the meaning unit was categorized under the domain for which 2 or all of the coders agreed. These discussions could also lead to a refinement of the coding strategy. After consensus on meaning units for all interviews, all meaning units were evaluated once more by one researcher (MMP) to generate collections of meaning units with similar underlying beliefs and this process was reevaluated by a second researcher (JWK) to confirm the belief statements.

Action taken:

We have rewritten the paragraph, which now states the following:

“We agreed on a coding approach to ensure consistency in the coding process. This approach encompassed that coding of the transcripts should be performed by reading the transcripts and searching for specific words or statements that were considered to be related to the contents of the 14 TDF domains as described by Atkins et al. [39]. Also, before analyzing the transcribed data, the coders examined the domains of the TDF and agreed on how to interpret and apply each domain. For example, it was agreed that the optimism domain would refer to the physicians’ expressed own optimism/pessimism and not that of other health care professionals or patients. Also, it was agreed that statements referring to aspects of physical time would be coded under the “Environmental Context and Resources” domain. After agreeing upon this coding approach, two of the researchers (JWK and MMP) independently coded the first two interview transcripts. This was done by reading the transcribed interviews and by considering if any meaning units (i.e. “a constellation of words or statements that relate to the same central meaning”[46]) that were related to the target behavior were of relevance to any of the TDF domains. Meaning units of relevance to the TDF domains were then allocated to the relevant domain [39]. The allocations of meaning units were discussed between coders and possible differences were considered until consensus was reached, after which the coding strategy was refined [39]. This process (i.e. coding, consensus, refinement) was used for all interviews”.

Comment 12:

 Line 162- please explain what you mean by process supervisor

Reply to comment 12:

We thank the reviewer for pointing out the lack of clarity in the term “process supervisor”. We refer to PN, who has considerable experience with using the TDF. He acted as a supervisor concerning questions that emerged regarding the framework and the analysis process.

Action taken:

We have changed the wording to indicate that PN supervised the analysis process.

“…who acted as a supervisor during the analysis process to ascertain that the application of the TDF was satisfactory”.

Comment 13:

Line 168: more information is needed about what the coding strategy was and how it was used

Reply to comment 13:

The coding strategy was an approach we used in the sense that the coders agreed on how to apply the TDF to the data set. The three coders discussed the 14 domains of the TDF and how these should be interpreted when applied in this study. For example, it was agreed that optimism referred to the physicians’ optimism or lack hereof and not the optimism/pessimism of other staff or patients’. Also, it was agreed that meaning units about time would be coded under “environmental context and resources”.

Action taken:

We have rewritten the entire paragraph and have changed the wording to:

“We agreed on a coding approach to ensure consistency in the coding process. This approach encompassed that coding of the transcripts should be performed by reading the transcripts and searching for specific words or statements that were considered to be related to the definitions of the 14 TDF domains as described by Atkins et al. [4]. Also, before analyzing the transcribed data, the coders examined the domains of the TDF and agreed on how to interpret and apply each domain. For example, it was agreed that the optimism domain would refer to the physicians’ expressed own optimism/pessimism and not that of other health care professionals or patients. Also, it was agreed that statements referring to aspects of physical time would be coded under the “Environmental Context and Resources” domain. After agreeing upon this coding approach, two of the researchers (JWK and MMP) independently coded the first two interview transcripts. This was done by reading the transcribed interviews and by considering if any meaning units (i.e. “a constellation of words or statements that relate to the same central meaning”[5]) that were related to the target behavior were of relevance to any of the TDF domains. Meaning units of relevance to the TDF domains were then allocated to the relevant domain [4]. The allocations of meaning units were discussed between coders and possible differences were considered until consensus was reached and hereby the coding strategy was refined [4]. This process (i.e. coding, consensus, refinement) was used for all interviews.”

Comment 14:

Line 171: please provide a more detailed description of what is a meaning unit and how these align with the TDF. How did the team identify the meaning units? Looks like constant comparative analysis was used to differentiate meaning units- is this what was done? Please provide more detail about your coding strategy- what was it?

Reply to comment 14:

We have clarified what we mean by a meaning unit by using the definition by Graneheim and Lundman in their paper Qualitative content analysis in nursing research: concepts, procedures and measures to achieve trustworthiness, Nurse Education Today (2004) 24, 105–112, which states that at meaning unit is “a constellation of words or statements that relate to the same central meaning”.

Action taken:

We have added this definition to the manuscript.

Comment 15:

Line 176: what was your percent agreement between coders? Did MMP serve as the gold standard or expert that the other coders were compared to?

Reply to comment 15:

The percentage agreement between coders is presented in the Intercoder Agreement paragraph in the results section in which we have written the following:

“In the first 3 interviews, 126 meaning units were allocated to one of the 14 domains. Initial agreement was found on 63 % of the codes by the two coders, JWK and MMP. After consensus discussion, agreement was obtained for 83 % of the codes, which were kept for further analysis, and agreement on removal of codes was obtained for 17 % of the codes, which were only identified by one coder and which were considered not to be related to the target behavior (Table 1). In the last 4 interviews, 160 meaning units were assigned to one of the 14 domains. Initial agreement on coding was found on 37 % of the codes by the two coders, MMP and RB. After consensus discussion, agreement on coding was obtained for 79 % of the codes, and agreement on removal of codes was obtained for 21 % of the codes (Table 1).”

MMP did not serve as the expert, but was the only coder performing initial coding of all interviews and was therefore used as the comparator. JWK was the expert and had previous experience with the TDF. Therefore, JWK and MMP compared their coding of the first interviews, discussed their differences in coding until consensus was reached and created a coding strategy to guide the following codings. Hereafter, all three coders coded the following three interviews and compared their codes, where after the two unexperienced coders (MMP and RB) performed the initial coding of the last seven interviews. JWK participated in the consensus discussions of all interviews.

Comment 16:

The whole section that contains information about how meaning units were used is very confusing and hard to follow impacting the reviewer’s ability to understand what was done and if the analysis adhered to rigorous standards. This section needs improvement

Reply to comment 16:

We thank the reviewer for pointing out this lack of clarity in the manuscript. We have followed the guide put forward by Atkins et al (Atkins et al. Implementation Science (2017) 12:77).

Action taken:

We have re-written the paragraph and hope that the process is described more clearly now.

“After agreeing upon this coding approach, two of the researchers (JWK and MMP) independently coded the first two interview transcripts. This was done by reading the transcribed interviews and by considering if any meaning units (i.e. “a constellation of words or statements that relate to the same central meaning”[46]) that were related to the target behavior were of relevance to any of the TDF domains. Meaning units of relevance to the TDF domains were then allocated to the relevant domain [39]. The allocations of meaning units were discussed between coders and possible differences were considered until consensus was reached, after which the coding strategy was refined [39]. This process (i.e. coding, consensus, refinement) was used for all interviews. The following three interviews were coded independently by all three coders, whereas the remaining seven interviews were coded by two coders (MMP and RB). To ensure transparency, we calculated agreement between the coders’ initial allocation of meaning units for the first 3 interviews (JWK and MMP) and the last four interviews (MMP and RB). This was done by simple percentage agreement [47]. After the initial coding of all interviews, differences in meaning unit allocations were discussed by all three researchers until negotiated consensus was reached on distribution of all meaning units. When a meaning unit was relevant to more than one domain of the TDF, it was coded into several domains. In some cases, all assigned domains for one meaning unit were agreed upon, and in other cases negotiated consensus was reached on de-assigning a domain to a given meaning unit. The main reason for de-assigning domains was that the meaning unit was not related to the target behavior [39].”

Comment 17:

Line 183- using the phrase “frequently reported” is too quantitative for a qualitative study. How often you count something does not equate with importance- major flaw that needs to be addressed. I am concerned the team is quantifying qualitative data or forcing quantitative principles into a qualitative study. Being able to generate more codes could simple be due to how the question was asked which could stimulate more discussion. Not generating as many codes could also be a problem with how the question was asked which may have shut down responses from participants.

Reply to comment 18:

This is a valuable comment. We are well aware that counting codes is a controversial approach to analyzing textual data, as stated by Morgan [1]. We have used counts as a way of discovering patterns in the transcribed material and to get a sense of which topics, of course guided by the interviews, were most consistently raised by the physicians. We do acknowledge that the use of “frequently reported” can be misleading and have omitted these words from the manuscript.

We agree with the reviewer that how often you count something does not equate with its importance and that the way the interviews were conducted may have encouraged/discouraged elaboration of answers. Therefore, we did include this limitation in our submitted manuscript in the discussion.

“Thirdly, we chose to focus on the determinant domains covering at least 5 % of the meaning units. However, it is possible that the most frequently identified barriers and facilitators may not be the barriers and facilitators of highest importance. Thus, there is a risk that some significant barriers and facilitators were omitted [7]. Additionally, the extensive interview guide may have made it difficult to obtain in-depth information since the guide may steer the interviewer away to new questions instead of allowing sufficient time for the informant to really explore each question”.

Action taken:

None

Comment 18:

 Line 191: what is meant by “belief statements”? what are these

Reply to comment 19:

Again, we thank the reviewer for pointing out this lack of clarity.

Action taken:

We have re-written the section for clarification:

“One researcher (MMP) reviewed the meaning units within each domain to generate collections of meaning units with similar underlying beliefs[4,5]. For each collection of meaning units, a belief statement was generated. A belief statement was defined as a statement with a specific underlying belief [4]. Thus, within each of the theoretical domains all meaning units were grouped with meanings units reflecting the same underlying belief [4]. Also, opposing meaning units that were considered to reflect opposing views on the same overarching belief were grouped together. For example, under the “Beliefs about consequences” domain these responses “I think it’ll be the effect of a physician mentioning it”, and “Well, I have to say, if I talk about it, being a physician, then I think that for a busy nurse it will be prioritized a little more. Or maybe it’ll put a focus on remembering mobility” were grouped under the belief statement “Patients and staff will focus more on mobility if the physician mentions mobility. The frequency of occurrence of meanings units within each belief statement across all interviews was counted. Another researcher (JWK) subsequently reviewed the beliefs statements to confirm these”.

Comment 19:

Line 192-193 are you referring to a core or main category?

Reply to comment 19:

We hope that we have clarified, as mentioned above. We are referring to statements with similar underlying beliefs and have re-written the paragraph for clarification.

Action taken:

Please see “reply to comment 18”.

Comment 20:

 Ethics section: did members of the research team have a working relationship with physician participants? If so, this needs to be disclosed. How did the team address confidentiality of the data?

Reply to comment 20:

As stated in the ethics section, we ensured full anonymity of each physician throughout the process. All interviews/participants were assigned and ID-number and at no time during the analysis and reporting of the data, the participants’ names were used. The link between ID and participant was stored on a closed computer drive to which only the three of the authors had access and this was of storing and handling data was approved by the Danish Data Protection Agency.

Action taken:

We have added this statement “None of the research members had a working relationship with any of the participating physicians”

Comment 21:

Intercoder agreement: again really confused about what is a meaning unit- is this a label assigned? Also it is always a struggle when quantitative methods are being forced into a qualitative study- as a reviewer telling me % agreement leaves many more questions. It looks like codes were simply tossed out- did the research team return to the data where analysts did not agree and recode together occur? Based on % it appears that 50% of codes were tossed out!

Reply to comment 21:

Meaning unit is a label assigned. As added to the manuscript, a meaning unit is “a constellation of words or statements that relate to the same central meaning”.

We are sorry to see that the reviewer feels that quantitative methods are being forced into our qualitative study. We do not see it this way, but have added the quantitative aspect for transparency, and we hope that the reviewer will agree to consider it this way.

We must have expressed ourselves unclearly and hope that the revised manuscript appears clearer to the reviewer. We did return to the data where we did not agree to make a consensus re-evaluation of the meaning unit in question. So recoding was performed together. This is also stated in the manuscript “After the initial coding of all interviews, differences in meaning unit allocations were discussed by all three researchers until negotiated consensus was reached on distribution of all meaning units. When a meaning unit was relevant to more than one domain of the TDF, it was coded into several domains. In some cases, all assigned domains for one meaning unit were agreed upon, and in other cases negotiated consensus was reached on de-assigning a domain to a given meaning unit. The main reason for de-assigning domains was that the meaning unit was not related to the target behavior [4]”.

Also, we thank the reviewer for pointing out that our data seemed to indicate that 50% of the codes were tossed out. This was definitely not the case, and we discovered an error in our Table 1 regarding the percentages. The percentages in Table 1, are relative to the number of meaning units analyzed for agreement. Agreement was only calculated for interviews 1-3 and 9-12, and the calculated agreement is therefore not an absolute number reflecting agreement across all interviews. For interviews 1-3, the agreement after consensus discussion was 105/126=83% and agreement after consensus for interviews 9-12 was 127/160=79.4%. Based on interviews 1-3 and 9-12, 54 out of 286 (21%) of the initially assigned meaning units were de-assigned based on consensus and primarily because the units were not related to the target behavior. Thus, consensus was reached on 79 percent of the units analyzed for agreement, which was based on half of the interviews.

We do acknowledge that the reviewer, and do agree with the reviewer, that calculating agreement may leave (and leaves) many more questions, but it also illustrates that despite analyzing transcripts meticulously based on an agreed strategy, analysis is not clear cut and straight forward and requires more than one coder to ensure that the data are treated as holistically as possible.

Comment 22:

 Line 228: prior you identified a total of 286 (126 from 3 interviews and 160 from the last 4 interviews) and in this line you indicate that there were 411 meaning units- which is it? This needs to be clarified

Reply to comment 22:

Thank you for this comment. As stated in the answer above, the identified meaning units reported in Table 1 (N=286) were from interviews 1-3 and 9-12. In total, based on all interviews, we identified 411 meaning units.

Action taken:

None.

Comment 23:

Table 3: there is a difference between awareness (know of the intervention) and knowledge- how are you differentiating between these 2 constructs

Reply to comment 23:

We do not differentiate between the 2 constructs. In “A guide to using the Theoretical Domains Framework of behaviour change to investigate implementation problems” by Atkins et al., the domain is “Knowledge (an awareness of the existence of something)” and covers the constructs Knowledge (including knowledge of condition/scientific rationale), Procedural knowledge and Knowledge of task environment. Therefore, we chose to consider knowledge and awareness to be covered by the knowledge domain.

Action taken:

None

Comment 24:

Table 3: social/professional role and identity- the section on bedrest- is the participant referring to cultural understanding among physicians that being ill means staying in bed or cultural understanding of patients that being ill means staying in bed- if it is a cultural understanding among patients- what does this have to do with social/professional role identify?

Reply to comment 24: 

This is a very valid question. It is actually a cultural understanding among patients, relatives and staff, which challenges the physicians’ professional role/identity, which is why these statements were put under “Social/professional role”.

Action taken:

None

Comment 25:

Table 3: belief statement the cooperation between physicians and physio is non-verbal: the quote provided does not speak to cooperation between physicians and physiotherapist- it indicates there is an assumption that work will be done. The second quote also does not support cooperation- the physician disregards the physio contributions!

Reply to comment 25: 

The reviewer is right in pointing out this mistake. It was a copy-paste mistake, which we have corrected by replacing the second statement by one of the statements expressing this meaning.

Action taken:

We have inserted this statement in Table 3:

We actually do not communicate that much with the physiotherapists. Sometimes, we prescribe physiotherapy or occupational therapy” (Physician no. 4, Department X)”

Also, we have changed the title of the meaning unit to better reflect the statements – “cooperation” has been changed to “communication”.

Comment 26:

Results: in this section there is reference to “themes”. There is much variation in how Themes are worded and how they are applied. But in general, and most scholars agree, that Themes should represent the abstract that brings meaning and identity to a participants experience and captures the wholeness of data that falls within the theme. Sometimes the authors state belief statements and other times they state themes within the apriori identified domain- this is very confusing. When theme is used is represents the data superficially or rather presented as a statement- for example line 338-339: “The intervention will cause improved cross-disciplinary cooperation”. This is not stated as an abstraction. I suggest the research go back and review how they are using the concept Theme in the results section.

Reply to comment 26:

Thank you for capturing this mistake. It is a typo and should have been “belief statement”.

Action taken:

Core theme” has been replaced by “belief statement”.

Comment 27:

Discussion: the authors continue to present results in the discussion section lines 479-504.

Reply to comment 27:

The reviewer is right in this comment and we agree that the information is misplaced.

Action taken:

We have moved the information from lines 479-504 to the results section.

Reviewer 2 Report

Thank you for the opportunity to review this very interesting paper. The introduction is well written and the rationale for the study adequately justified. The methods are also clear although I do not have specific expertise in qualitative research methodology.

I do have some comments related to the presentation of the results and the discussion/ conclusion which are outlined below:

Results: 

Table 2- the percentage column has some commas instead of dots e.g., 1,46. Please correct. 

Lines 245-263: Is this supposed to be part of the table legend for Table 3? If so I think this is quite a lot of detail and is unnecessary. It is obvious from the information included in the table what 'opposing' views means. 

lines 264-464- I think this whole part of the results section is quite hard to read. There is too much detail. This makes it hard to appreciate the main themes and concepts that were elucidated from the structured interviews. It is useful to include some illustrative quotations but these should be restricted as much as possible to maintain the flow of the text. Otherwise it is both repetitive (i.e., repeating information documented in Table 3) and clumsy, with the important messages lost in a sea of 'who said what when'. Please could the authors address this. 

Discussion and Conclusion:

lines 477-78: could the authors explain more clearly the phrase 'targeting multiple levels to influence physicians’ attitudes and behaviors' and explain  why their findings support this approach?

lines 479-504 are very repetitive of the results section and make the discussion sluggish. I think these paragraphs could be summarised more succinctly and re-phrased. The discussion should contextualise the results, explaining what the study has added and how the information gathered fits with what was already known. Subsequent paragraphs do this more successfully and so I think this part of the discussion could be shortened/ merged with subsequent sections, to make the manuscript more succinct. 

It would also be helpful for the reader to understand more about what future work is needed to build on the results presented here. How will this study help with designing an intervention to improve mobility in older hospitalised adults? This may become clearer if the authors provide a better explanation of the phrase 'targeting multiple levels to influence physicians' attitudes and behaviors....'.  

Author Response

Dear Revuewer 2

Please find our reply to your comments below:

Comment 1:

Thank you for the opportunity to review this very interesting paper. The introduction is well written and the rationale for the study adequately justified. The methods are also clear although I do not have specific expertise in qualitative research methodology.

I do have some comments related to the presentation of the results and the discussion/ conclusion which are outlined below:

Reply to comment 1:

We thank the reviewer for these very positive words about our manuscript.

Comment 2:

Results: 

Table 2- the percentage column has some commas instead of dots e.g., 1,46. Please correct. 

Reply to comment 2:

Thank you for pointing out this mistake. It’s been corrected.

Action taken:

Commas replaced with dots in Table 2.

Comment 3:

Lines 245-263: Is this supposed to be part of the table legend for Table 3? If so I think this is quite a lot of detail and is unnecessary. It is obvious from the information included in the table what 'opposing' views means. 

Reply to comment 3:

No, this is not part of the table legend for Table 3, but part of the text body. We apologize for this mistake, which seems to have occurred in the upload of the manuscript. Lines 245.263 are part of the section “Key Domains”.

Action taken:

None.

Comment 4:

lines 264-464- I think this whole part of the results section is quite hard to read. There is too much detail. This makes it hard to appreciate the main themes and concepts that were elucidated from the structured interviews. It is useful to include some illustrative quotations but these should be restricted as much as possible to maintain the flow of the text. Otherwise it is both repetitive (i.e., repeating information documented in Table 3) and clumsy, with the important messages lost in a sea of 'who said what when'. Please could the authors address this. 

Reply to comment 4:

Thank you for making us aware of this repetitive and clumsy appearance of our results section.

Action taken:

We have addressed the reviewer’s comment by removing all illustrative quotations, which are mentioned in Table 3, from the text. We hope that this has made the test more reader friendly.

Comment 5:

 Discussion and Conclusion:

lines 477-78: could the authors explain more clearly the phrase 'targeting multiple levels to influence physicians’ attitudes and behaviors' and explain why their findings support this approach?

Reply to comment 5:

Thank you for the comment. We believe that we have explained this comment in lines 470-476:

These domains reflect both individual and more social influences on physicians’ attitudes and behaviors to achieve increased mobility in hospitalized older medical patients. Thus, whereas Knowledge, Optimism and Beliefs about consequences are associated with physicians’ individual cognitions, Social/Professional role and identity and Environmental context and resources are more related to external influences on physicians’ attitudes and behaviors, e.g. expectations on their behaviors in various work situations and circumstances in their work environment that discourage or encourage certain behaviors”.

Action taken:

We do acknowledge that this could be written more clearly and have changed 'targeting multiple levels to influence physicians’ attitudes and behaviors’ to 'targeting both internal (I.e. individual) and external levels to influence physicians’ attitudes and behaviors’

Comment 6:

lines 479-504 are very repetitive of the results section and make the discussion sluggish. I think these paragraphs could be summarised more succinctly and re-phrased. The discussion should contextualise the results, explaining what the study has added and how the information gathered fits with what was already known. Subsequent paragraphs do this more successfully and so I think this part of the discussion could be shortened/ merged with subsequent sections, to make the manuscript more succinct. 

Reply to comment 6:

The reviewer is right, indeed, that this is not rightfully part of a discussion.

Action taken:

We have moved the section to the results section. This has also been requested by reviewer 1.

Comment 7:

It would also be helpful for the reader to understand more about what future work is needed to build on the results presented here. How will this study help with designing an intervention to improve mobility in older hospitalised adults? This may become clearer if the authors provide a better explanation of the phrase 'targeting multiple levels to influence physicians' attitudes and behaviors....'.  

Reply to comment 7:

Thank you for this comment and for pointing to this lack of clarity. With “targeting multiple levels”, we refer to interventions being needed to address determinants at different system levels, from individual physicians (optimism, beliefs about consequences) and the profession (social/professional role and identity) to emergency departments and hospitals (social influences and environmental context and resources).

Action taken:

We have changed the wording in the conclusion to this:

These barriers reflect different system levels of influences on physicians’ behaviors, from the individual physician (optimism, beliefs about consequences) and the profession (social/professional role and identity) to the broader context in which the physicians work (environmental context and resources). Thus, it is recommended that future interventions aiming to increase mobility in hospitalized older medical patients should target multiple levels to influence physicians’ promotion of mobility”.

Round 2

Reviewer 1 Report

Comment 1: response- thank you for adding directed content analysis to the methods. I am a bit concerned with your reference, the Hsieh et al article which provides a nice overview of 3 different types of content analysis- pros and cons with brief examples of how coding can be used. In and of itself, is not an in-depth resource that details how to conduct a directed content analysis. However, you could cite that your use of frequency with appearance of codes aligns with Hsieh description.

Comment 10: if you followed Hsieh et al- they provide specific examples of how to code and analyze data- this is not evident in your manuscript- providing concerns about rigor of the analysis and if the process as described by Hsieh was followed- this needs to be adjusted and addressed in your manuscript

Author Response

Thank you for taking the time to review our manuscript. We have changed the manuscript according to the reviewer's suggestions and hope that we addressed the shortcomings sufficiently.

Comments from Reviewer 1;

Comment 1:

response- thank you for adding directed content analysis to the methods. I am a bit concerned with your reference, the Hsieh et al article which provides a nice overview of 3 different types of content analysis- pros and cons with brief examples of how coding can be used. In and of itself, is not an in-depth resource that details how to conduct a directed content analysis. However, you could cite that your use of frequency with appearance of codes aligns with Hsieh description.

Reply to comment 1:

We thank the reviewer for this comment. We have reviewed the reference and have added, as suggested by the reviewer, that the use of frequency is inspired by Hiesh and Shannon.

Correction:

In line 205, we have written:

Therefore, inspired by Hsieh and Shannon [44] we considered the frequency of meaning units assigned to a domain as a measure of its importance at the time of the interviews.

Comment 10:

if you followed Hsieh et al- they provide specific examples of how to code and analyze data- this is not evident in your manuscript- providing concerns about rigor of the analysis and if the process as described by Hsieh was followed- this needs to be adjusted and addressed in your manuscript

Reply to comment 10:

We thank the reviewer for pointing to this lack of clarity. We have tried to re-write the data-analysis paragraph (lines 163-211) to comply with the reviewer’s request and hope that we have clarified the process sufficiently.

Correction:

The wording of the paragraph was changed to:

We conducted a qualitative directed content analysis with a focus on manifest content [42–44]. According to Joffe and Yardley, manifest content refers to what is directly observable and visible in the data [43]. Three researchers (JWK, MMP and RB) carried out the analysis. One of these researchers (JWK) had previous experience in using the TDF with another researcher (PN) [45], who acted as a supervisor during the analysis process to ascertain that the application of the TDF was satisfactory. The aim of the analysis was to identify the most common barriers and facilitators among the physicians concerning the WALK-Cph mobility intervention and their own promotion of in-hospital mobility. We defined barriers as physicians’ negative attitudes and behaviors in relation to the intervention and/or their promotion of mobility, while facilitators were defined as physicians’ positive attitudes and behaviors concerning the intervention and/or their promotion of mobility.

We analyzed the interviews using a directed content analysis as described by Hsieh and Shannon [44] and deductively analyzed the transcribed interviews using an iterative process based on the TDF [39]. Firstly, we agreed on a coding approach to ensure consistency in the coding process. This approach encompassed that coding of the transcripts should be performed by reading the transcripts and searching for specific words or statements that were considered to be related to any of the contents of the 14 TDF domains, as described by Atkins et al. [39]. Each domain of the TDF was considered a coding category [44]. Secondly, the coders examined the domains of the TDF and agreed on how to interpret and apply each domain. For example, it was agreed that the optimism domain would refer to the physicians’ expressed own optimism/pessimism and not that of other health care professionals or patients. Also, it was agreed that statements referring to aspects of physical time would be coded under the “Environmental Context and Resources” domain.

After agreeing upon this coding approach, two of the researchers (JWK and MMP) independently coded the first two interview transcripts. Each coder independently read each interview and identified and highlighted meaning units (i.e. statements) that were related to the target behavior. Then the two coders re-read the interviews and considered if any meaning units (i.e. “a constellation of words or statements that relate to the same central meaning”[46]) that were related to the target behavior were of relevance to any of the TDF domains. Meaning units of relevance to the TDF domains were then allocated to the relevant domain [39].

After independent coding of the interviews, the allocations of meaning units were discussed between all three coders and possible differences were considered until consensus was reached, after which the coding approach was refined [39]. This process (i.e. coding, consensus, refinement) was used for all interviews. The subsequent three interviews were coded independently by all three coders, whereas the remaining seven interviews were coded by two coders (MMP and RB). After the initial coding of all interviews, differences in meaning unit allocations were discussed by all three researchers until negotiated consensus was reached on distribution of all meaning units. When a meaning unit was relevant to more than one domain of the TDF, it was coded into several domains. In some cases, all assigned domains for one meaning unit were agreed upon, and in other cases negotiated consensus was reached on de-assigning a domain to a given meaning unit. The main reason for de-assigning domains was that the meaning unit was not related to the target behavior [39]. Finally, to ensure transparency, we calculated agreement between the coders’ initial allocation of meaning units for the first 3 interviews (JWK and MMP) and the last four interviews (MMP and RB). This was done by simple percentage agreement [47].

Reviewer 2 Report

The authors have addressed my initial comments

Author Response

Thank you for taking the time once more to read our manuscript. We're happy to see that we have addressed all comments sufficiently.